# Concept and Design of a Multi-Polarization Reconfigurable Microstrip Antenna with Symmetrical Biasing Control

**DOI:** 10.3390/s24082408

**Published:** 2024-04-10

**Authors:** Filipa Antunes, Amélia Ramos, Tiago Varum, João N. Matos

**Affiliations:** 1Department of Electronics, Telecommunications and Informatics, University of Aveiro, Campus Universitário de Santiago, 3810-193 Aveiro, Portugal; filipa.antunes23@ua.pt (F.A.); ameliaramos@ua.pt (A.R.); matos@ua.pt (J.N.M.); 2Instituto de Telecomunicações, Campus Universitário de Santiago, 3810-193 Aveiro, Portugal

**Keywords:** reconfigurable antennas, polarization reconfigurable, microstrip antenna

## Abstract

Wireless communication systems have grown rapidly, moving towards being highly compact, intelligent, and flexible to adapt to changing operating requirements. Multifunctional and highly versatile antennas are key in this development to ensure system quality. Reconfigurable antennas, particularly regarding polarization, allow frequency reuse and enable the mitigation of fading effects. This work presents a square microstrip patch antenna operating in the ISM 5.8 GHz band with reconfigurable polarization by controlling its feeding. This antenna has four different states through the application of a symmetrical DC voltage that controls an RF circuit with PIN diodes. As a result, the microstrip patch can operate with three different polarizations: linear polarization and both circular polarizations (right-handed and left-handed). The antenna was fabricated to validate the proposed concept. The good agreement between the measurement and the simulation results was possible to observe regarding its polarization behaviour, impedance adaptation and radiation pattern.

## 1. Introduction

Wireless communications are integral to modern society and have been instrumental in technological advancements over recent decades. The proliferation of wireless technologies presents numerous challenges, particularly in integrating diverse systems to better serve people and users. Practically everywhere, there are sensors and different devices integrating communication networks or even sharing different networks in the same space. Antennas serve as essential components in all devices utilizing wireless communication networks, whether they are sensors or fixed/mobile terminals. They act as interfaces between guided electronic signals and free space, significantly influencing communication capacity and quality. Antennas are characterized by their operating band, radiation pattern, and polarization, among other metrics. In most cases, different wireless systems share the same communication medium but operate with different characteristics, whether it is the working frequency, polarization, or coverage, and consequently, with different antennas. As wireless communication evolves, there is a growing need for antennas that can adapt to changing parameters. In the future, a single sensor or terminal may be used by different applications, requiring antennas that are capable of adjusting their characteristics accordingly and adapting to this variation in parameters, becoming reconfigurable.

A reconfigurable antenna entails the ability to alter one of its inherent characteristics while preserving the same physical structure. In summary, reconfigurability allows a single antenna to accomplish multiple specifications, making it suitable for different types of systems. Some of the most extensively researched attributes in this field include frequency variation [1], radiation pattern alterations [2], and polarization change.

Reconfigurable polarization offers several advantages, including adaptability to different environments, polarization diversity (which minimizes signal fading), and frequency reuse. This is particularly valuable in crowded spectrum environments, as it enables a more efficient use of frequency bands [3]. In the context of this work, polarization will be the focus. Polarization is the orientation of the electric field in space. The polarization can be classified as linear, circular, or elliptical [4], and it is consistent for both transmitter and receiver antennas. The various basic radiating structures have in common the fact that they essentially produce linear polarization. However, typically, the polarization can be changed in these basic antennas (enabling other linear, circular, or even elliptical polarizations in the most general case) mainly through modifications in the physical structure of the antenna or in the antenna’s feeding. The solution through antenna feeding, however, presents greater versatility and opens a vast set of opportunities to explore. Several works have already been presented in the literature exhibiting antennas with some reconfigurability in terms of polarization.

In [5,6,7,8,9,10,11,12,13,14], a number of antennas with polarization reconfiguration through changes in the radiating structures were reported. Regarding square-shaped patches or their variants, several antennas are presented in [5,6,7,8,9,10,11] with switching possibilities between linear and circular polarizations (RHCP/LHCP), for instance, in [5], where four PIN diodes were used along with four metal walls, making the structure complex and difficult to construct. The structure in [6] operates in three different polarizations controlled by six PIN diodes, conditioning the current distribution in the antenna. One of the negative aspects of this structure is that the wires that control the diodes disturb the radiation area. Using two PIN diodes to control a slot structure, a radiating solution based on a microstrip patch with a single feed that switches between linear and circular polarization is presented in [7]. These types of structures have narrow polarization bandwidths.

An E-shaped patch structure that allows switching between left and right circular polarizations is shown in [8], and it uses two RF switches that are PIN diodes. Three capacitors are still used to block the DC signal, making the radiating element complex. Another possibility is presented in [9] with a patch whose four corners are cut and connected using four independent PIN diodes, and by controlling them, the generation of different linear and circular polarizations is possible. In [10], the authors present another alternative that generates three different polarizations (linear and LHCP/RHCP) by adjusting extra segments in the corners of a square patch. These segments are connected using PIN diodes. In [11] an aperture-coupled patch is presented. It generates ±45° polarizations using four RF switches in a cross-shaped aperture that creates two distinct diagonal slots. These solutions that change the physical structure of the radiating element have as their main problems the conditioning of the radiating part with soldered elements, increasing the complexity of the structure but also leading to reduced bandwidths.

In [12,13], two structures based on circular patches are presented. In [12], the structure is based on a ring with two switches and six non-metallic columns that provide three different polarization states: linear and circular LHCP and RHCP, while in [13], it is shown a circular patch using four switches that are PIN diodes, enabling four distinct linear polarizations: H/V and ±45°. These solutions present a structure that is difficult to build and quite bulky. The authors in [14] present a printed square monopole with two strips in the ground plane that are connected using two PIN diodes. Depending on whether the diodes are on or off, the monopole operates three different polarizations: linear as well as circular LHCP/RHCP. This antenna does not present boresight radiation, in addition to the problems of reduced bandwidth and high sensitivity of performance to variations in the ground plane strips.

There are other structures seen in [15,16,17,18,19] that generate different polarizations by working on the feeding part of the antennas. The authors in [15] present a reconfigurable antenna with four polarization states, combining a single feed to generate ±45° linear polarization and a dual feed to obtain LHCP/RHCP circular polarization. To do so, they use eight PIN diodes in the architecture, showcasing a bulky antenna structure with a parasitic patch above a radiating ring, as well as six control voltages. In [16], a structure is based on a circular patch fed at two points 90° out of phase using a T-shaped divider and four graphene nanoplates connected in open circuit lines. By applying a certain DC voltage, the resistance of the nanoplates varies, allowing the current flow in such a path, turning the ports on and off, and generating vertical and horizontal polarization as well as RHCP.

Four different polarizations (two orthogonal linear and two orthogonal circular) are obtained in [17] using a feeding network developed to generate these four different states using eight switches, and it feeds the square path at two points. This antenna uses a feed network with some complexity using slow-wave coupled lines. Another solution was proposed in [18], where a feed network connected to a circular patch at four different points uses eight PIN diodes to obtain four different linear polarizations with an interval of 45° (22.5°, 67.5°, 112.5° and 157.5°). A reconfigurable multi-polarization antenna is proposed in [19] with the ability to switch between four different linear polarizations with 45° intervals using a circular patch antenna with four shorting posts. By controlling the connection of each of the four shorting posts with the ground plane using PIN diodes, the antenna can generate linear polarizations of 0°, 45°, 90° and 135°. Both structures produce only linear polarizations.

The literature shows that with the proper feeding network, basically by double-feeding an antenna orthogonally, it is possible to efficiently generate different polarizations. In this work, by feeding a square patch into two orthogonal points, it was possible to develop a four-state feeding network that allows the antenna to generate three types of polarization using a single bias voltage.

This paper is divided into four different sections, starting with an introduction presenting the framework and revealing the state of the art. The second section presents the architecture of the antenna and the feeding network. In the third section, the prototypes developed are presented and characterized with their respective simulation and measured results. Finally, the last section presents the conclusions of this work.

## 2. Antenna Architecture and Design

In this work, microstrip antennas were used due to their simplicity, low cost, low profile, and compatibility with printed circuits. They are also structures that present great versatility for the design of antennas, with the possibility of generating different characteristics. A microstrip patch antenna consists of a dielectric substrate between two conductive layers, typically copper. One layer is completely filled and serves as the antenna’s ground plane, while the other, only with a portion of copper, operates as the radiating structure [20]. Microstrip patches are typically designed to operate in TM10 or TM01 modes, depending on the alignment of the feed point on the patch (with an *x* or *y*-axis). These modes produce broadside radiation patterns [21]. By default, a microstrip patch antenna (with a rectangular or circular shape) with a single feed and no structural alterations radiates linear polarized electromagnetic waves (horizontal or vertical, according to the feeding point) [4]. It is, however, possible to obtain other types of polarization (a) through modifications to the physical structure of the patch or (b) through feeding the patch using two orthogonal points [4]. The structure proposed in this work is based on this latter technique, as it allows a variety of different polarizations depending on the signals that feed the antenna at these two orthogonal points. In this configuration, the TM01 and TM10 modes are excited in the patch simultaneously.

Considering a square microstrip patch as a standalone radiating structure, applying the aforementioned technique with the feeding of two orthogonal points (rotated 90°) through microstrip transmission lines leads to the structure shown in Figure 1.

If the patch were singularly fed by either one or the other branch independently, it would radiate electromagnetic waves with linear (vertical or horizontal) polarization. In this work, both branches were fed simultaneously, and by working with them, it will be possible to generate multiple polarizations for the antenna.

By feeding each point with the same amplitude of signals and controlling the phase delay between the different branches, different polarizations can be generated. In the case of feeding at both patch points with signals in phase, the polarization generated will be linear with a 45° slant, while when the phase shift is ±90°, the particular case of generating circular polarization (left or right, depending on the sign) occurs.

As a result, to reach these different polarization states, one needs to introduce different phase delays, and that can be accomplished simply by varying the electrical length of the transmission lines that feed the antenna. By controlling the phase in the feeding at two orthogonal points of a square microstrip antenna, this work aims to obtain three different types of polarization: (i) linear 45° slanted, (ii) left circular and (iii) right circular. The proposed concept is shown in Figure 1, where a transmission line feeding path is presented, and later, using a basic T-junction power splitter, it is divided into two sub-paths (Left Pathway and Right Pathway) that go towards two points that feed the patch on two orthogonal faces, and each of the main sub-paths is called “Main Segment Left” and “Main Segment Right”. Additionally, two sections of line were added, with an electrical length greater than the Main Segment by 90°, which are called “Phase Segment Left” and “Phase Segment Right”.

By controlling which of the paths (either “Main” or “Phase”) the signal will undergo in each Pathway, it will be possible to generate the three different polarizations in the antenna. In this sense, it was necessary to develop a biphasic switching circuit control.

The antenna and transmission lines were designed using microstrip technology printed on a single-layer PCB and to operate at 5.8 GHz central frequency. The Isola Astra MT77 was the selected dielectric substrate, with a thickness of 0.76 mm, a relative permittivity of ɛ_r_ = 3.00, and a loss tangent tan δ = 0.0017 @ 10 GHz.

### 2.1. Symmetric Bias-Controlled Biphasic Phase Shifter

Figure 2 presents the layout of the designed biphasic 0°/90° Phase Shifter (PS) device that is based on switched delay lines [22]. Reconfigurability in this context refers to the ability to control at each moment which of the pathway lines will be active (Main or Phase), and, therefore, which polarization the antenna has at that moment. The blue line (Phase Segment) has an additional electrical length of λ/4 in comparison to the red line (Main Segment); therefore, the signal phase offset between the two segments corresponds to 90°.

This circuit, in addition to the two segments connected to the common RF 100 Ω characteristic impedance transmission lines, also includes four PIN diodes (D1, D2, D3, and D4), arranged in two pairs with opposite orientations. At any given moment, one pair of diodes connects (in forward-biased mode) one segment whereas the other pair isolates the other segment (in reverse-biased mode). Table 1 summarizes the different states for each diode in the circuit that turn each segment ON or OFF. OFF represents reverse bias, whereas ON means forward bias.

The PIN diode used was the MA4SPS402 [23], which has as its main characteristics a series resistance of R_s_ = 5 Ω, and a total capacitance C_t_ of 0.045 pF. There is still a biasing circuit (Grey colour) that connects both segments to the ground (for signal), allowing the switching of the diodes to be conducted using a symmetric voltage ±Vin. Considering Vin = 4.8 V, two equal resistances R1 = R2 = 390 Ω were also added to the polarization circuit to balance the operation of the two diodes simultaneously and ensure a forward current of I_f_ = 10 mA. DC coupling capacitors (C1 and C2) were also integrated into the PS electronic circuit. After the design process, the biphasic switching circuit was included in the architecture previously presented (Figure 1).

### 2.2. Multi-Polarization Reconfigurable Microstrip Patch Antenna

The biphasic switching circuit was introduced into the middle of each left and right Pathways, and the resulting antenna architecture is shown in Figure 3. The antenna structure includes a square patch fed on two orthogonal faces through a double quarter-wavelength transformer to match its input impedance to the assumed characteristic impedance for the transmission lines, 100 Ω. On the other side of the phase switching circuit, there is a basic power divider, T-Junction, that connects two 100 Ω lines to a simple input line with a 50 Ω characteristic impedance. All relevant dimensions are presented in Table 2.

The final antenna has four operation modes that depend on the values of Vin1 and Vin2 and whether these are equal to +Vin or −Vin. A summarized version of how to attain the different polarization states is presented in Table 3, as well as the segments that are connected to each pathway for each mode. As previously mentioned, the antenna was designed assuming Vin = 4.8 V. It is important to mention that the LP1 and LP2 modes correspond to the same linear polarization (45° slant), since the patch is fed with equal amplitude and in phase.

Figure 4 shows, for the different operating states of the designed antenna, the variation of the simulated electric field along the patch, for four fixed moments during a period T. For example, for the CP1 state it is possible to see that for the instant t = 0 s, the feeding signal at the vertical point (bottom side of the patch) has a value close to zero, while at the horizontal point (right side of the patch) the feeding signal has a high value, and the field distribution (instantaneously) resembles the TM01 mode. After a quarter of the period t = T/4 s the instantaneous situation is the opposite, that is, the feeding signal at the vertical point presents a maximum while at the horizontal point it presents a minimum, and in this case the instantaneous image of the distribution of the electric field resembles TM10 mode. It should be noted that in the case of states CP1 and CP2, the antenna feed signals are out of phase by T/4 s from each other to enable the generation of circular polarization.

For the CP2 state, there are some similarities with the field figures from the CP1; however, with a time delay of T/4. These field distributions vary in time since the TM10 and TM01 modes are always excited simultaneously, as mentioned before. In the case of states LP1 and LP2, both orthogonal feeding signals are in phase, and it is possible to observe in both states, at each instant, that in the two points the maximums (and the minimums) coincide, generating a field distribution with a slope of 45°.

Another analysis that can be performed concerns surface currents. Figure 5 presents the distribution of simulated surface currents for the four states over four instants during a period T. According to Figure 5, it is possible to see that in the case of the CP1 state, the direction of the currents rotates to the left side over time, proving that the polarization generated is left circular. In state CP2, the behaviour is similar, although in this case it is possible to observe that the direction of variation is opposite, and therefore, in this state, right circular polarization is generated. For the states LP1 and LP2, the surface current vectors vary with an inclination of 45°, demonstrating the generation of linear polarization.

## 3. Results and Discussion

The structures of the biphasic circuit control and the final antenna were fabricated and will be presented and characterized in this section.

### 3.1. Phase Shifter

The device for the phase shifter was manufactured, and the prototype is shown in Figure 6.

The circuit was measured using a Vector Network Analyser, and the result of the S_21_ parameter over frequency is presented in Figure 7, in terms of the input/output attenuation (Figure 7a) and the input/output phase shift (Figure 7b). Both Figures present the results relating to the Main Segment and Phase Segment, that is, Vin = +4.8 V and Vin = −4.8 V, respectively.

Observing a, it is clear that the attenuation is higher when the Phase Segment path is active compared to when the Main Segment is active. This is expected given the longer path length of the Phase Segment, which results in greater losses. In fact, at 5.8 GHz, attenuation is around 0.24 dB and 0.04 dB, respectively, meaning that the device presents overall very low insertion losses at the working frequency. Regarding the phase difference, the measured phase when the Phase Segment is active is 38°, and when the Main Segment is connected, the phase is −50.7°. In conclusion, the phase switch circuit operates as intended, presenting an 88.7° input/output phase shift at 5.8 GHz between the two ±Vin states.

### 3.2. Multi-Polarization Reconfigurable Microstrip Patch Antenna

The designed polarization reconfigurable antenna was manufactured and assembled, and the prototype is shown in Figure 8.

The antenna’s reflection coefficient was measured, and the S_11_ parameter for the four different possible polarization states is shown in Figure 9, where a comparison with the respective simulations is presented. Analysing Figure 9, the first aspect that stands out is the slight deviation in the frequency between the group of measured results and the simulated ones. This can be justified by (i) misalignments in the manufacturing/assembly process and/or (ii) possible impacts of the parasitic elements of the non-linear devices in the structure. However, it can be clearly seen that the structure is resonant around the frequency of interest for the four states. In fact, for the different states, the minimum values of the measured reflection coefficient are −11 dB, −18 dB, −20 dB and −23 dB for LP1, CP2, CP1, and LP2, respectively, at frequencies around 5.85 GHz. Assuming the commonly used criterion of S_11_ < −10 dB, measured bandwidths greater than 110 MHz can be observed for states CP1, CP2 and LP2.

Figure 10 presents the comparison between the simulated and measured results of the Axial Ratio in the direction of boresight for the four possible states. High values of axial ratio denote that one component of the polarization ellipse is much higher than the other, revealing a polarization that tends to be linear. On the opposite side, low values of axial ratio suggest that the axes of the polarization ellipse are approximated, revealing a polarization that tends to be circular.

It is possible to verify that states relating to linear polarization LP1 and LP2 present high axial ratio values, while on the other hand, for CP1 and CP2, which comprise states with circular polarization, low values of axial ratio are shown. The slight frequency deviation between simulations and measurements that was observed in the S_11_ result can also be seen in the axial ratio; nevertheless, it is possible to validate the various polarization states. According to Figure 10, the minimum measured AR values around 5.85 GHz are 1.3 dB and 6.5 dB for CP1 and CP2, respectively, revealing a circular polarization in the antenna. Even though in the CP2 state, the axial ratio presented a higher value than in simulations, one cannot exclude the possibility that the minimum axial ratio in this state is in a nearby direction and not at boresight. For LP1 and LP2, the measured values are 16.5 dB and 41 dB, both high values, validating the 45° slanted polarization.

Figure 11 presents the comparison between simulation and measurements of the various normalised radiation patterns of the antenna at 5.85 GHz in the two main planes Φ = 0° and Φ = 90° for the four polarization states of the antenna. For simplicity of analysis, the states with circular polarization present curves with the left and right components of the electric field, E_LHCP_ and E_RHCP_, while the states related to linear polarization present the co-polar and cross-polar components, E_co-pol_ and E_cross-pol_.

The first aspect that is important to highlight is that in all eight sub-figures, there is a great approximation between the simulated and the measured results. Going into more detail about the different polarizations, Figure 11a,b show the two planes Φ = 0° and Φ = 90° for the CP1 state of the antenna. It is possible to verify that this state corresponds to left-hand circular polarization, which throughout the radiation pattern is dominant in relation to the right component, and in the boresight direction, it presents a measured rejection between the two components greater than 16 dB in both planes.

Regarding the CP2 state, it is possible to observe in Figure 11c,d that for both planes, the polarization remains circular but changes direction to become right-hand circular polarization since the right component overlaps the left component throughout the radiation pattern. Only a small degradation in the measured component can be noted in the Φ = 0° plane, which had already been seen in the axial ratio value. This outcome could be related to some misalignment in the manufacturing or measurement process, since the observations in the plane Φ = 90° show a good rejection between components already around 15 dB.

Figure 11e,f correspond to the antenna LP1 state, and Figure 11g,h are related to the LP2 state, which, although two different states of the antenna configuration, both correspond to the same 45° slant linear polarization since the patch antenna is fed on both sides with the same phase. In addition to the already mentioned good matching between the various simulated and measured curves, it is possible to verify a difference of around 20 dB for the cross-polarization component, which reveals a good linear polarization in the co-polar component. Finally, the measured gain of the reconfigurable antenna designed is 5.9 dBi in the CP1 state, 5.5 dBi in the CP2 state, 6.2 dBi in the LP1 state, and 6.7 dBi in the LP2 state, in line with the simulation’s expectations.

In Table 4, a comparison of the main characteristics between the designed antenna and the more relevant reported works related to multi-polarization-reconfigurable antennas is carried out. It can be observed that most works use PIN diodes as a switching element due to their simplicity and functionality; however, the proposed work was the one (between those that use PIN diodes) that uses fewer bias control voltages to operate in the four states. In fact, in this work, a single pair of symmetric voltages is necessary. It is important to note that in this developed structure, compactness was not a requirement.

## 4. Conclusions

In this work, a multi-polarization reconfigurable antenna printed on a PCB was developed, consisting of a dual orthogonal feeding microstrip patch antenna with a controlled phase shifting circuit between two paths through a symmetrical DC voltage. In the four operation modes (CP1, CP2, LP1, LP2), an S_11_ below −10 dB was achieved, only with a slight deviation in the optimal operation frequency. In terms of axial ratio, the results validate the generation of linear and circular polarization in the different states. The radiation patterns showed a good match between the simulated and measured results, validating the generation of three distinct types of polarization in the antenna: left-handed circular, right-handed circular, and linear polarization 45° slanted, through the correct selection of the DC voltage used to control each of the pathways that feed the patch antenna.

## Figures and Tables

**Figure 1 sensors-24-02408-f001:**
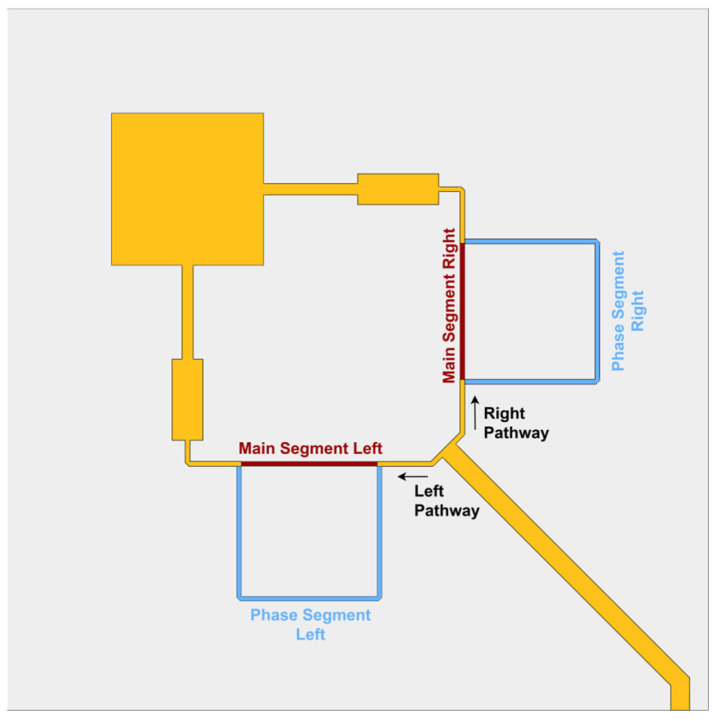
Concept of a dual-feed patch antenna with flexible polarization through different feeding phases.

**Figure 2 sensors-24-02408-f002:**
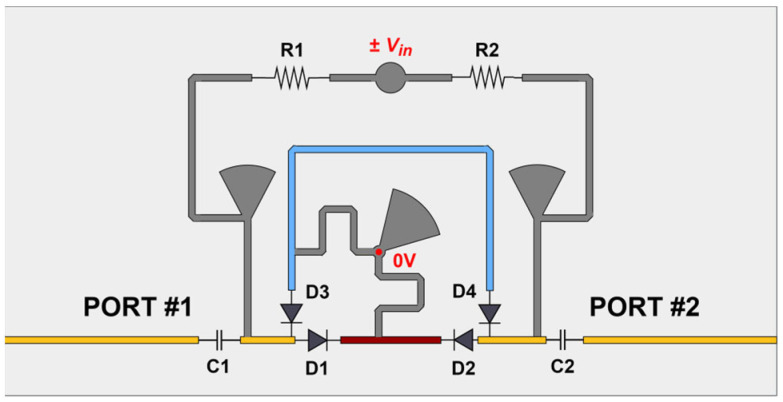
Layout of the designed dual-state 0°/90°, symmetrically bias-controlled phase shifter.

**Figure 3 sensors-24-02408-f003:**
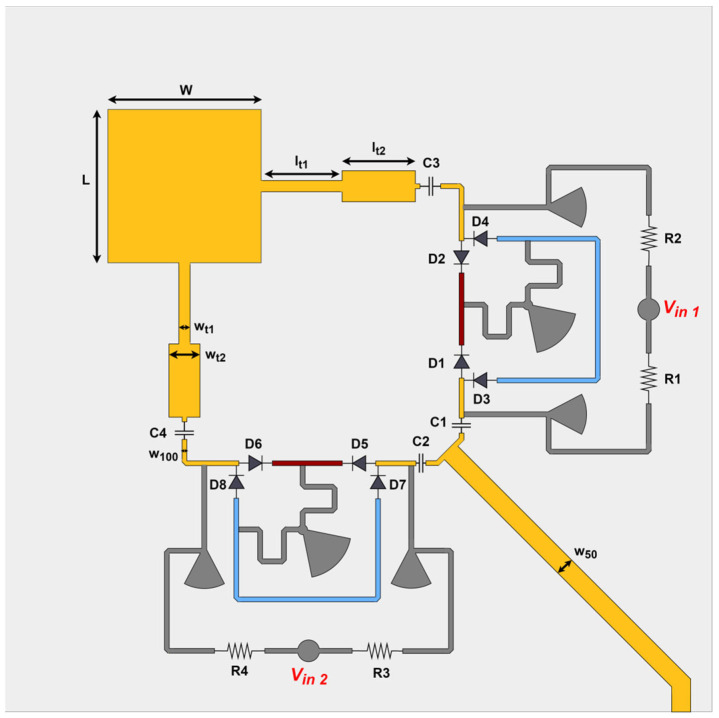
Layout of the multi-polarization reconfigurable patch antenna with three polarization states and four operation modes.

**Figure 4 sensors-24-02408-f004:**
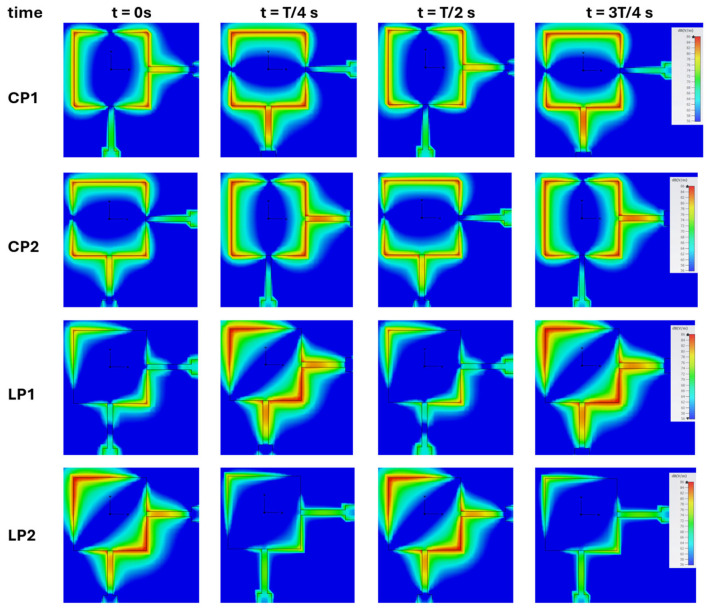
Variation of the electric field in the patch for four instants over a period T, for the four different operating states.

**Figure 5 sensors-24-02408-f005:**
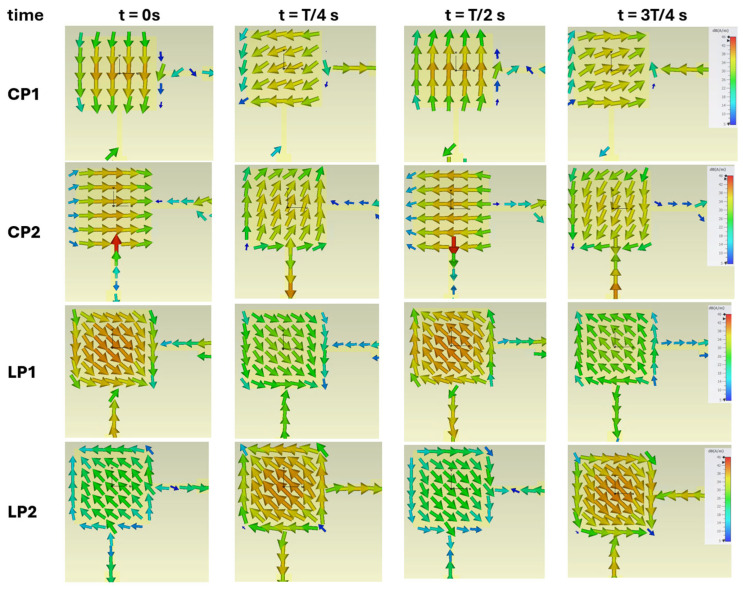
Variation of surface currents in the patch for four instants over a period T, for the four different operating states.

**Figure 6 sensors-24-02408-f006:**
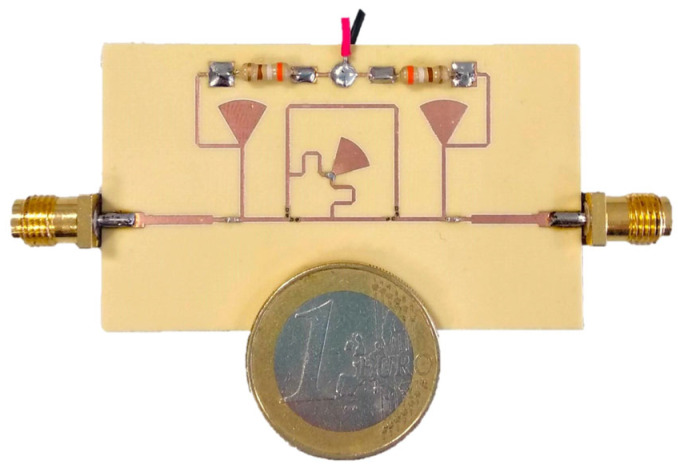
Photo of the fabricated phase-shifter prototype.

**Figure 7 sensors-24-02408-f007:**
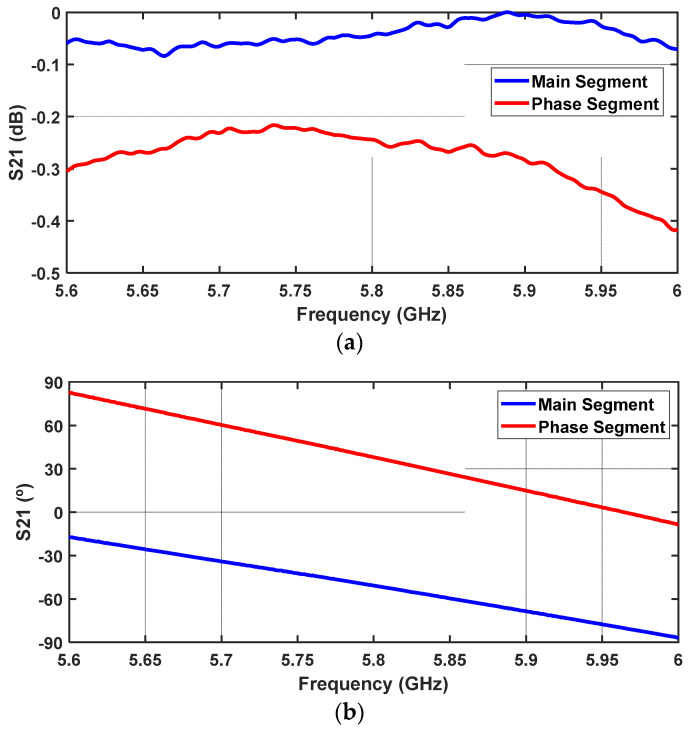
Measured S_21_ over frequency of the biphasic phase shifter: (**a**) magnitude of S_21_ input/output—Insertion loss of each line segment; (**b**) phase of S_21_—phase delay for each segment.

**Figure 8 sensors-24-02408-f008:**
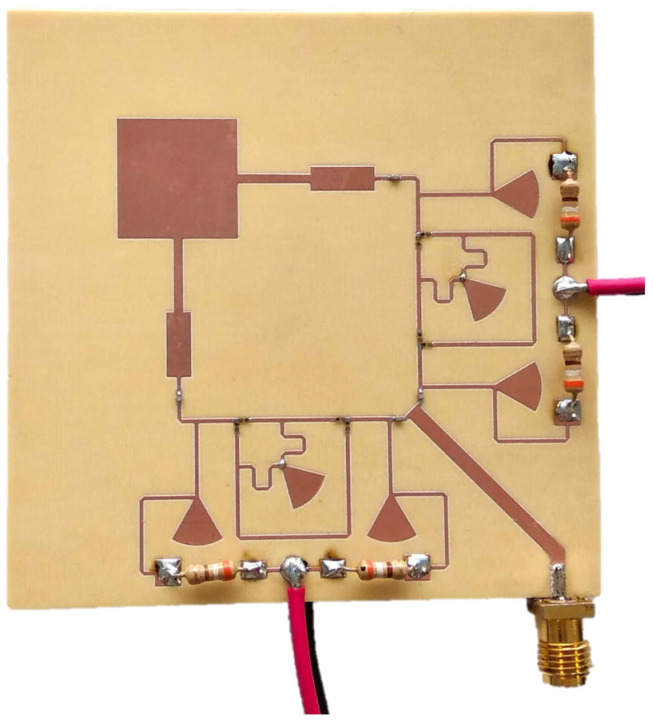
Photo of the fabricated multi-polarization reconfigurable microstrip antenna.

**Figure 9 sensors-24-02408-f009:**
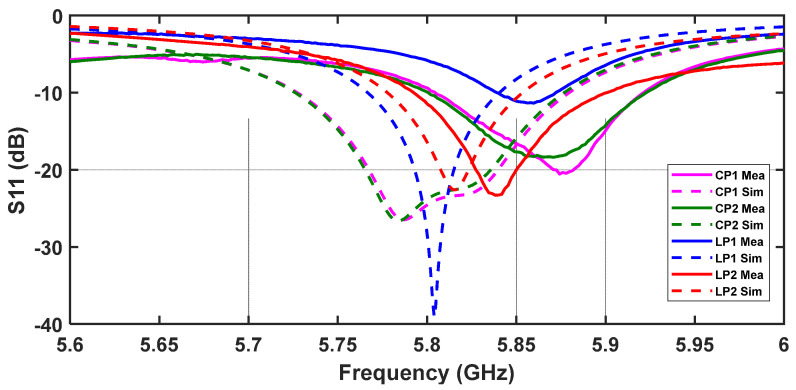
Comparison between the simulated and measured S_11_ of the designed polarization reconfigurable microstrip antenna for the four different polarization states.

**Figure 10 sensors-24-02408-f010:**
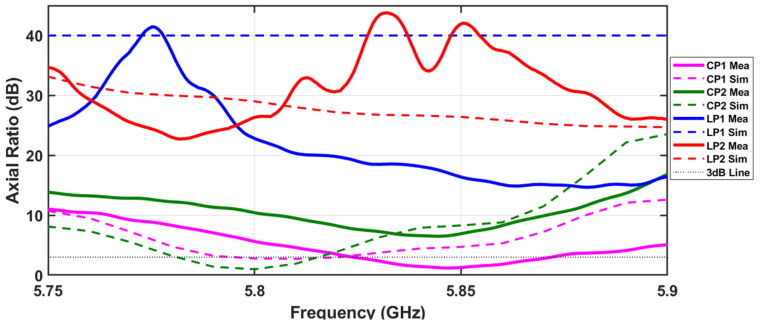
Comparison between the simulated and measured axial ratio of the designed polarization reconfigurable microstrip antenna for the four different polarization states.

**Figure 11 sensors-24-02408-f011:**
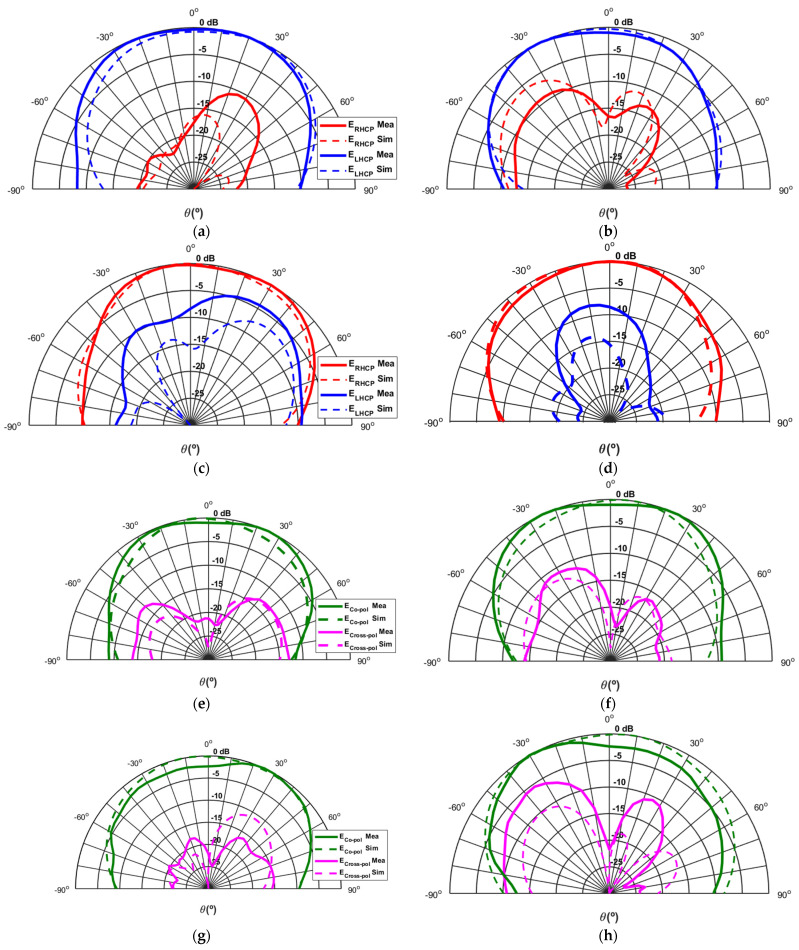
Comparison between simulated and measured radiation patterns of the designed polarization re-configurable microstrip antenna for the four different polarization states: (**a**) Polarization state CP1—radiation plane Φ = 0°; (**b**) Polarization state CP1—radiation plane Φ = 90°; (**c**) Polarization state CP2—radiation plane Φ = 0°; (**d**) Polarization state CP2—radiation plane Φ = 90°; (**e**) Polarization state LP1—radiation plane Φ = 0°; (**f**) Polarization state LP—radiation plane Φ = 90°; (**g**) Polarization state LP2—radiation plane Φ = 0°; (**h**) Polarization state LP2—radiation plane Φ = 90°.

**Table 1 sensors-24-02408-t001:** Forward and reversed biased diodes to connect each segment.

Microstrip Line Segments	Diodes	
	D1 and D2	D3 and D4
Main Segment	ON	OFF
Phase Segment	OFF	ON

**Table 2 sensors-24-02408-t002:** The resulting dimensions of the Multi-Polarization Reconfigurable Microstrip Antenna.

W = L	w_t1_	w_t2_	l_t1_	l_t2_	w_100_	w_50_
14.34 mm	1.04 mm	2.88 mm	8.79 mm	7.66 mm	0.44 mm	1.77 mm

**Table 3 sensors-24-02408-t003:** Antenna polarization states according to control V_bias_ applied for each phase shifter.

Mode	Right Pathway	Left Pathway	Vin1 (V)	Vin2 (V)
CP1	Phase Segment	Main Segment	−Vin	+Vin
CP2	Main Segment	Phase Segment	+Vin	−Vin
LP1	Main Segment	Main Segment	+Vin	+Vin
LP2	Phase Segment	Phase Segment	−Vin	−Vin

**Table 4 sensors-24-02408-t004:** Comparison between proposed and other reported reconfigurable polarization antennas.

Ref.	Antenna Type	Frequency (GHz)	Antenna Size (λ^2^)	Control Mechanism	No. of Switches	No. Control Voltages	No. States	Polarization States
[15]	Ring Patch	2.45	0.57 × 0.57	PIN Diodes	8	6	4	LHCP, RHCP, −45° LP, +45° LP
[16]	Circular Patch	10.85 11.85	NA	Graphene nanoplate	4	2	3	RHCP, HP, VP
[17]	Square Patch	0.93	0.25 × 0.25	PIN Diodes	8	4	4	LHCP, RHCP VP, HP
[18]	Circular Patch	2	NA	PIN Diodes	8	4	4	22.5° LP, 67.5° LP, 112.5° LP, 157.5° LP
[19]	Circular Patch	2.45	0.57 × 0.57	PIN Diodes	4	4	4	0° LP, 45° LP, 90° LP, 135° LP
**This work**	**Square Patch**	**5.8**	**1.35 × 1.35**	**PIN Diodes**	**8**	**2**	**4**	**LHCP, RHCP, +45° LP**

## Data Availability

Data is contained within the article.

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
