# Peer review of "Concept and Design of a Multi-Polarization Reconfigurable Microstrip Antenna with Symmetrical Biasing Control"

_sensors, 2024, doi:10.3390/s24082408_

Round 1

Reviewer 1 Report

Comments and Suggestions for Authors

The authors propose a Concept and Design of a Multi-Polarization Reconfigurable Microstrip Antenna with Symmetrical Biasing Control. A prototype is fabricated and experimentally verified as well. This reviewer finds teh work interesting but has a suggestion to make. Since different polarizations are excited by the configuration of the microstrip patch antenna, how are the modes of the patch changing? It is essential to find the modes as they will effect the radiation pattern. For ex- in [a],

[a] “A Novel half Hemispherical Dielectric Resonator antenna with array of slots loaded with a circular metallic patch for wireless applications”, AEU – International Journal of Electronics and Communication, Elsevier, Vol. 69, pp. 1755 – 1759, 2015

a circular patch is loaded on a half CDRA. Now, with the changing resonant modes, the mode of circular patch also changes which affect the radiation pattern of the whole antenna configuration. Similarly, in your proposed design how are the modes changing? Please assess with current distributions and claim your modes as well.

Comments on the Quality of English Language

It needs minor editing.

Author Response

The authors would like to thank the reviewer for all the review and the comments, which we found very relevant and touched on an aspect that unfortunately we had not given special focus to in the submitted paper. Thanks to the reviewer we were able to add this information in the document with more detail. As the proposed antenna is a square patch (symmetric structure) fed at two orthogonal points and, according to the different states presented, the phase of each feeding signal at these points is controlled, we can say that the two main modes of patch TM01 and TM10 are excited, and the field distribution changes in time according to the feeding phases, in the different states.

Figure 1 shows the variation of the electric field in the four operating states of the patch, over a period T, in four instants. It is possible to verify that, for the CP1 state, at the instant T=0s, the signal that feeds the patch at the vertical point has a minimum value, so at that moment the distribution of the electric field is similar to the typical TM01, after a quarter of a period the positions change, with the feeding point at the horizontal presenting the minimum value, which leads to the fact that at a time T/4, the field distribution resembles the TM10 mode. The same happens for the CP2 state. In cases LP1 and LP2 the two modes are excited in phase. We also added Figure 2 to the paper, which presents the surface currents and can reinforce the information regarding the polarization states, and visually conclude that in cases CP1 and CP2 the polarizations are circular, as a rotation in the directions of surface currents is visible in time, in the case of CP1 with a rotation to the left side over time, while in CP2 it is in the opposite direction. In cases LP1 and LP2 the polarization is the same, linear with an orientation at 45º, and it coincides with the direction of the surface current vectors.
(figures in the attached file)

Reviewer 2 Report

Comments and Suggestions for Authors

The paper is commendably written and presented, showcasing a thorough investigation into the development of a multi-polarization reconfigurable antenna printed on a PCB. The authors demonstrate a comprehensive understanding of the subject matter and effectively communicate their findings. However, there are minor changes that, if implemented, could further enhance the quality and acceptability of the paper.

Here are some suggested corrections and improvements to the paper:

  1. Clarify and simplify the opening sentence:

"Wireless communications are integral to modern society and have been instrumental in technological advancements over recent decades."

  1. Streamline the explanation of wireless technologies and their challenges:

"The proliferation of wireless technologies presents numerous challenges, particularly in integrating diverse systems to better serve people and users."

  1. Break down complex sentences for better readability and understanding:

"Antennas serve as essential components in all devices utilizing wireless communication networks, whether they are sensors or fixed/mobile terminals. They act as interfaces between guided electronic signals and free-space, significantly influencing communication capacity and quality. Antennas are characterized by their operating band, radiation pattern, and polarization, among other metrics."

  1. Improve transition to discussing reconfigurable antennas:

"As wireless communication evolves, there is a growing need for antennas that can adapt to changing parameters. In the future, a single sensor or terminal may be used by different applications, necessitating antennas capable of adjusting their characteristics accordingly."

  1. Clarify the concept of reconfigurable polarization and its advantages:

"Reconfigurable polarization offers several advantages, including adaptability to different environments, polarization diversity (which minimizes signal fading), and frequency reuse. This is particularly valuable in crowded spectrum environments, as it enables more efficient use of frequency bands."

6.       Break down the paragraph discussing various structures and solutions for reconfigurable antennas for better readability and comprehension.

7.       In line 235-236, the results are not well explained and illustrated.  “Observing Figure 5 a) it is possible to verify that the attenuation is higher when the 235 Phase Segment path is active than when the Main Segment is active, which is expected 236 given the longer path length (and the losses associated).

8.       The conclusion accurately summarizes the key findings and implications of the study on the multi-polarization reconfigurable antenna but these points may enhance it:

1.       Functionality Verification: Check if the conclusion accurately reflects the functionality of the antenna. Does it indeed describe how the antenna can generate multiple polarization states (circular and linear) through controlled phase shifting and voltage selection?

2.       Provide a comparison table of the results between the paper findings and previous literature including: antenna type, polarization states, control mechanism, frequency range , validation and application.

3.       Technical Details: Check if the conclusion provides sufficient technical details about the antenna design, such as the use of a dual orthogonal feeding microstrip patch antenna, controlled phase shifting circuit, switch delay line technique, and PIN diodes.

4.       Overall Implications: Consider the broader implications of the study's findings. Does the conclusion highlight the significance of the multi-polarization reconfigurable antenna for practical applications or further research in the field?

9.       Ensure that the citations follow a consistent format style throughout the document.

Comments on the Quality of English Language

See general comments.. 

Author Response

The authors would like to thank the reviewer for his careful work in reviewing this article, and for all the suggestions that were appreciated, accepted, and based on them allowed the authors to improve the document in the new version now submitted.
1.to  6. - The authors thank the reviewer for his suggestions for improving the document's writing, we found them all relevant, we accepted them, and based on them we tried to change the document, and we think it is now better and clearer.
7.-Thank you very much for the point, in the new version of the document we adapted the text in the description of the figure.
8.-The authors understand the reviewer's suggestion to improve the conclusions, and based on them we changed the document. We have added the comparison table at the end of the document, and we have also tried to adjust the conclusions. Thank you so much again.
9.-Thanks to this point, we analyzed all the citations in the document and corrected any aspect that required modification.
All changes are reflected in the newly submitted version.

Reviewer 3 Report

Comments and Suggestions for Authors

This paper designs a patch antenna with Polarization Reconfigurable function, this work is based on conventional idea and design, some comments as follow:

1. This work lacks method novelty. The basic idea of this work, by using orthogonal feeding structure with 0/90deg phase difference to change polarization of the antenna is a ordinary method in antenna design.

2. This work lacks design novelty and technical contributions. both patch antenna and the feeding structure is conventional structure. 

3. In the introduction, the reference works are listed, but unable to see the advantages of this design and the technical issues it aims to address.

4. In fig.8, Since the feeding structure is symmetrical, the simulated AR curves of CP1 (purple) and CP2 (green) should be overlap. It show a over 3dB difference between the two curves. This indicates some simulation errors.

5. In fig.8, For the linear polarization results, significant differences can be seen between the simulation and measurement results, without any explaination.

6. In fig.9, For the CP2 results (fig.9c and 9d), the 0deg points in (c) and (d) are defined in the same space direction, They are the normal direction of the patch antenna. This means the results must be the same in 0deg points. This indicates some measurement errors.

Taking the above comments into consideration, this paper must be rejected.

Comments on the Quality of English Language

The quality of english is about the average. the expression need to be more concise and logical, some expression must be accurate.

Author Response

Regarding points 1 and 2, the authors understand the reviewer's idea, however we would like to clarify the innovative aspects. If we look at the state-of-the-art references, the majority use orthogonal feeding, and still they are relevant works published recently in reference journals in the area, an aspect which showcases the great interest in these structures. The authors would like to highlight that in this work, more than dual-orthogonal feeding, we were able to obtain three distinct polarizations in a patch antenna using a symmetric bias voltage, unlike most other works. Moreover, that was achieved in a very simple way that, we believe, is based on the concept presented which is still scientifically relevant. Thanks again to the reviewer.

Regarding point 3, we understand the issue raised, and in the new version, we tried to improve the description of the state of the art.

Regarding Point 4, the authors presented the results as they were obtained by simulation. We would like to mention that the upper conductive structure of the antenna is symmetrical in relation to the Patch itself, but there are slight physical differences, either in the position of the input line or in the distance between the microstrip lines of the phase shifters and the substrate delimitations. Additionally, the simulation method used in the CST electromagnetic simulator, the Finite Integration Technique, creates a mesh for later use in simulation estimations. We are therefore unable to guarantee that there are no errors due to the mesh, and therefore the simulation, however, we believe that the main objective is to prove and validate the proposed technique. However, as the reviewer suggests, we believe that the discrepancies are substantially due to simulation errors. In fact, the Axial ratio is calculated through the ratio between radiated field components, so these discrepancies are later reinforced in subsequent calculations, such as the axial ratio.

Also, regarding Point 5, we do not hide the differences between simulations and measurements. In fact, we already note this difference in the document, but these differences observed do not invalidate the proof of the concept proposed. The PCB where the antenna is built was made internally at our workplace, meaning that we did not use any specialized company, and therefore we believe that one of the sources of error is the manufacturing process, which truth be told, does not have the precision that exists in specialized companies. A second (and important) source of error is the differences in the properties of the dielectric substrate, between the typical values used in the simulator and the value in the real substrate sheet. Lastly, and not least, there is the effect of the parasitic components of the PIN diodes, as well as their soldering and position in the circuit, so despite the deviation of around 50 MHz (around 8%), it was possible to verify the behavior of the antenna at the frequency of operation.

Finally, regarding Point 6, we analyzed the process, and we admit that there was an error in the measurement process, possibly in the rotation of the antenna (in the mechanical positioner) there may have been a physical misalignment, meaning that the measured values did not correspond in the two planes at the point in common. In this way, we once again performed new measurements in the CP2 state, taking more care with the alignment issue, which we updated in the document. We would like to thank the reviewer once again for his review, and for having detected issues that allowed us to clarify, correct and improve the document.

Round 2

Reviewer 3 Report

Comments and Suggestions for Authors

1. This article lacks innovation. It is based on conventional antenna structures and methods for antenna polarization realization. While the authors show some innovation in the design of the antenna feeding network, the design structure of this feeding network is complex and occupies a large size, which is not a very good design. However, in terms of scientific research, it does achieve the reconstruction of antenna polarization.

2. The author emphasizes in the response the errors in simulation and measurement, stating that these errors do not affect the conclusions of the antenna design. Maybe this is not a responsible attitude, although minor measurement errors are acceptable. As far as I know, in terms of simulation, CST is based on an improved FDTD method, and this software is already very mature. Under normal simulation settings, its simulation accuracy is completely reliable.

In conclusion, even if the presented simulation and measurement results do not affect the proof of the conclusions, misleading readers due to unreasonable errors may occur. In scientific research, while the proof of the conclusions is important, the rigor of the process data is even more crucial because it can reflect the scientific nature of the entire study.

Comments on the Quality of English Language

some expressions can be more concise

Author Response

Answer attached in pdf file.
